# Estimated impact of tafenoquine for *Plasmodium vivax* control and elimination in Brazil: A modelling study

Narimane Nekkab[1]*, Raquel Lana[2], Marcus Lacerda[3,4], Thomas Obadia[1], André Siqueira[5], Wuelton Monteiro[3,6], Daniel Villela[2], Ivo Mueller[1,7,8], Michael White[1]*

1 Malaria: Parasites and Hosts, Department of Parasites and Insect Vectors, Institut Pasteur, Paris, France, 2 Programa de Computação Científica, Fundação Oswaldo Cruz, Rio de Janeiro, Brazil, 3 Diretoria de Ensino e Pesquisa, Fundação de Medicina Tropical Dr. Heitor Vieira Dourado, Manaus, Brazil, 4 Instituto Leônidas e Maria Deane, Fundação Oswaldo Cruz, Manaus, Brazil, 5 Instituto Nacional de Infectologia Evandro Chagas, Fundação Oswaldo Cruz, Rio de Janeiro, Brazil, 6 School of Health Sciences, Universidade do Estado do Amazonas, Manaus, Brazil, 7 Population Health & Immunity Division, Walter and Eliza Hall Institute of Medical Research, Parkville, Victoria, Australia, 8 Department of Medical Biology, University of Melbourne, Melbourne, Victoria, Australia

* narimane.nekkab@pasteur.fr (NN); michael.white@pasteur.fr (MW)

**Data Availability Statement:** The model code is publicly available online at https://github.com/MWhite-InstitutPasteur/Pvivax_TQ_IBM.

## Abstract

### Background

Despite recent intensification of control measures, *Plasmodium vivax* poses a major challenge for malaria elimination efforts. Liver-stage hypnozoite parasites that cause relapsing infections can be cleared with primaquine; however, poor treatment adherence undermines drug effectiveness. Tafenoquine, a new single-dose treatment, offers an alternative option for preventing relapses and reducing transmission. In 2018, over 237,000 cases of malaria were reported to the Brazilian health system, of which 91.5% were due to *P. vivax*.

### Methods and findings

We evaluated the impact of introducing tafenoquine into case management practices on population-level transmission dynamics using a mathematical model of *P. vivax* transmission. The model was calibrated to reflect the transmission dynamics of *P. vivax* endemic settings in Brazil in 2018, informed by nationwide malaria case reporting data. Parameters for treatment pathways with chloroquine, primaquine, and tafenoquine with glucose-6-phosphate dehydrogenase deficiency (G6PDd) testing were informed by clinical trial data and the literature. We assumed 71.3% efficacy for primaquine and tafenoquine, a 66.7% adherence rate to the 7-day primaquine regimen, a mean 5.5% G6PDd prevalence, and 8.1% low metaboliser prevalence. The introduction of tafenoquine is predicted to improve effective hypnozoite clearance among *P. vivax* cases and reduce population-level transmission over time, with heterogeneous levels of impact across different transmission settings. According to the model, while achieving elimination in only few settings in Brazil, tafenoquine rollout in 2021 is estimated to improve the mean effective radical cure rate from 42% (95%

Restrictions apply for SIVEP malaria case reporting data. SIVEP data is owned by the Ministry of Health of Brazil which can be attained by meeting data access criteria by contacting the Health System Informatics Department (DATASUS) at datasus@saude.gov.br. Demographic data is publicly available at https://www.ibge.gov.br/. All other relevant data for modeling calibrations are within the manuscript and its Supporting Information files.

**Funding:** NN, MW, DV, and RL received funding from the Medicines for Malaria Venture (https://www.mmv.org/). NN has been in part supported by the Pasteur-Roux-Cantarini Fellowship by Institut Pasteur (https://research.pasteur.fr/en/call/pasteur-roux-cantarini-fellowship_call-for-application_2019-autumn-session/). NN, IM, and MW are supported by the Supporting Preparedness in the Asia-Pacific Region through Knowledge (SPARK) project under the ASEAN-Pacific Infectious Disease and Response program (APIDDaR) (https://www.spark.edu.au/). IM is supported by NHMRC Principal Research Fellowship (GNT1155075) (https://www.nhmrc.gov.au/funding/find-funding/research-fellowships). DV is a CNPq/Brazil research fellow (https://www.gov.br/cnpq/pt-br). The funders had no role in study design, data collection and analysis, decision to publish, or preparation of the manuscript.

**Competing interests:** I have read the journal's policy and the authors of this manuscript have the following competing interests: "NN, MW, DV, and RL received funding from Medicines for Malaria Venture (MMV) to evaluate the impact of roll-out of tafenoquine in Brazil."

**Abbreviations:** G6PDd, glucose-6-phosphate dehydrogenase deficiency; UI, uncertainty interval.

uncertainty interval [UI] 41%–44%) to 62% (95% UI 54%–68%) among clinical cases, leading to a predicted 38% (95% UI 7%–99%) reduction in transmission and over 214,000 cumulative averted cases between 2021 and 2025. Higher impact is predicted in settings with low transmission, low pre-existing primaquine adherence, and a high proportion of cases in working-aged males. High-transmission settings with a high proportion of cases in children would benefit from a safe high-efficacy tafenoquine dose for children. Our methodological limitations include not accounting for the role of imported cases from outside the transmission setting, relying on reported clinical cases as a measurement of community-level transmission, and implementing treatment efficacy as a binary condition.

## Conclusions

In our modelling study, we predicted that, provided there is concurrent rollout of G6PDd diagnostics, tafenoquine has the potential to reduce *P. vivax* transmission by improving effective radical cure through increased adherence and increased protection from new infections. While tafenoquine alone may not be sufficient for *P. vivax* elimination, its introduction will improve case management, prevent a substantial number of cases, and bring countries closer to achieving malaria elimination goals.

## Author summary

### Why was this study done?

- Radical cure with tafenoquine plus chloroquine has been recently approved to treat *P. vivax* malaria; however, the impact of improving individual-level effective radical cure —by overcoming non-compliance with primaquine with this single-dose regimen—on population-level transmission is unknown.

- Depending on age, pregnancy and lactating status, and glucose-6-phosphate dehydrogenase (G6PD) phenotypic activity and drug metabolism, treatment eligibility and effective radical cure rates may vary across different populations with variable transmission.

### What did the researchers do and find?

- With mathematical modelling, we accounted for these complex dynamics to consider the non-linear dynamics of primaquine and tafenoquine treatment on the burden of *P. vivax* malaria across a range of settings and for various implementation strategies.

- To our knowledge, our work is the first to show how rolling out tafenoquine in populations will improve case management of *P. vivax*, reduce transmission, and prevent a substantial number of cases, even in settings with high rates of effective case management with primaquine. However, tafenoquine alone will not lead to *P. vivax* elimination.

**What do these findings mean?**

- Accounting for the factors that lead to a variable degree of impact, our results can guide countries considering introducing tafenoquine with G6PD deficiency testing as part of tailored intervention strategies adapted to their local context.

- Overall, tafenoquine will have several public health benefits; however, it should be considered as an additional tool along with other interventions to reach elimination goals in the proposed time frame.

## Introduction

*Plasmodium vivax* malaria has proven to be particularly challenging to control with conventional malaria control measures [1]. In co-endemic regions, as malaria transmission levels decrease, *P. vivax* is able to persist better than *P. falciparum*, frequently becoming the dominant malaria species [2–6]. This resilience to control is largely attributable to its ability to relapse from long-lasting liver-stage hypnozoites, causing relapsing blood-stage infections weeks to years after the primary infection. Targeting the hypnozoite reservoir responsible for approximately 80% of all blood-stage infections is a key aspect of any strategy tackling *P. vivax* malaria [7,8].

At present, primaquine is the only widely available drug capable of eliminating liver-stage hypnozoites that is routinely used in case management [9]. However, its complex treatment schedule makes adherence a challenge, and its effectiveness is often less than optimal [10]. A new radical cure regimen with tafenoquine plus chloroquine has recently been licensed. In phase III trials, it demonstrated non-inferiority with respect to low-dose primaquine to prevent relapse whilst requiring the administration of only a single dose [11].

There are substantial challenges when extrapolating the results of clinical trials to estimate the potential impacts of primaquine and tafenoquine when administered to real populations. First, both pose the risk of potentially serious episodes of haemolysis when administered to individuals with glucose-6-phosphate dehydrogenase deficiency (G6PDd), an X-linked inherited blood disorder [12]. Tafenoquine is counter-indicated for individuals with below 70% G6PD activity, as determined by a quantitative G6PD test [13]. Therefore, eligibility of patients for tafenoquine will depend on the genotypic prevalence of G6PDd in populations, the G6PD activity levels of those with *P. vivax* infections, and the performance of the diagnostic tests [14]. Second, in the absence of directly observed therapy, adherence to a full course of primaquine treatment for either 7 or 14 consecutive days can be highly variable in real populations [15,16]. Third, the efficacy of liver-stage drugs can be undermined by insufficient dosing and low CYP2D6 metabolism [17]. Treatment failures due to low CYP2D6 metabolism have been observed for primaquine, while the impact on tafenoquine remains uncertain [11,18–21].

Mathematical models of *P. vivax* transmission can provide guidance and projections on a number of key areas surrounding the impact of introducing tafenoquine for first-line treatment of symptomatic cases. Such studies are particularly relevant for populations expecting to roll out tafenoquine in the coming years, such as in Brazil, which became the first endemic country to approve tafenoquine in October 2019 [22]. Despite readily available case management with primaquine provided free of charge by the government, Brazil has observed a rise in the number of reported malaria cases since 2016 [23]. With some surveys observing less than

70% adherence to the short-course primaquine regimen [24,25], Brazil's adoption of tafenoquine with quantitative G6PD testing may provide a new tool for controlling the recent surge of *P. vivax* malaria and progressing towards elimination.

In our study, we assess the impact of tafenoquine introduction, with Brazil as a case study of pre-existing primaquine case management practices, using a mathematical model of *P. vivax* transmission [26]. The data-driven modelling approach allows us to assess the impact on *P. vivax* transmission of prescribing tafenoquine instead of primaquine in eligible groups, assuming various levels of G6PDd prevalence, primaquine adherence, prevalence of low CYP2D6, and drug efficacy. We compare several distinct tafenoquine introduction scenarios, and assess the sensitivity of parameter estimates, to provide an overview of the potential impact of various treatment strategies.

## Methods

### Overview

Treatment pathways for radical cure of *P. vivax* infections were incorporated into a previously developed transmission model to better understand the potential impact of introducing tafenoquine into case management practices on population-level transmission dynamics (Fig A in S1 Text) [26]. Quantitative G6PD diagnostic testing was implemented in the model using Gaussian mixture models for men and women fitted to data from studies reporting phenotypic G6PD activity, calibrated to G6PDd genotype prevalence from a large Brazilian survey (Fig D and Fig F in S1 Text). We accounted for misclassification with the SD Biosensor G6PD test as a reference (Fig B in S1 Text). We evaluated the differential impact of varying 8-aminoquinoline eligibility criteria and efficacy, and conducted sensitivity analyses on adherence and prevalence of low CYP2D6 metabolism. Different scenarios of tafenoquine rollout in 2021 were simulated for municipalities with active transmission to evaluate the public health impact up to the year 2030. The treatment scenarios to be modelled were specified in advance of epidemiological data acquisition.

### *P. vivax* transmission model

We adapted a previously published mathematical model of *P. vivax* transmission to model case management with tafenoquine in Brazilian settings [26]. Fig A in S1 Text shows a compartmental representation of the transmission model. The model is implemented in 2 complementary formats: first, as a deterministic compartmental model described by a system of ordinary differential equations, and, second, as an individual-based stochastic model. Notably, the 2 model implementations provide identical predictions for large population sizes, when stochastic effects average out. The population dynamics of multiple mosquito species are described using compartmental models. The model was further adapted to reflect the epidemiology of *P. vivax* in Brazil by incorporating 2 distinct modes of transmission (peri-domestic and occupational) (see S1 Text for details). The model code is available online at https://github.com/MWhite-InstitutPasteur/Pvivax_TQ_IBM.

### Parameter estimates for the Brazilian setting

We used data from 2018 case reports from the Malaria Epidemiological Surveillance Information System (SIVEP-Malaria) database to estimate the incidence (light-microscopy-positive slides or positive rapid diagnostic tests reported per 1,000 population) for each municipality and to extract the eligibility criteria for treatment among cases (age, pregnancy, and lactation status) [27]. The model was calibrated for 126 Brazilian municipalities reporting at least 100

cases from January to December 2018 using case notification data from SIVEP-Malaria for incidence and occupational exposure risk, the Brazilian Institute of Geography and Statistics (IBGE) databases for population size and mean age, and a large genotyping survey for G6PDd prevalence (Fig E, Fig F, and Table D in S1 Text). We assumed the reported incidence represented the majority of clinical cases in the population. Parameters for treatment efficacy, low CYP2D6 metabolism, and G6PDd diagnostics were assumed to be the same across the municipalities and were estimated from the literature (S1 Text). For the Brazilian setting, we assumed an average relapse rate of 1/69 days$^{-1}$ based on work by Daher et al. [28]. Model calibration details are provided in Table A in S1 Text. Ethical approval was not required for our study.

Since several municipalities cover a large spatial area and population, we included an additional peri-urban setting to model variations between urban and rural transmission. Since the municipality of Manaus accounts for a population of over 2 million residents, with a large malaria-free urban area, we accounted for the intra-municipality heterogeneity in transmission by only modelling the peri-urban area of Manaus, informed by data from a longitudinal cohort [29]. Reported aggregate results include only peri-urban Manaus.

### Intervention scenarios

Using transmission levels from 2018, we compared a baseline scenario with the current radical cure regimen with chloroquine and primaquine to several scenarios for tafenoquine case management. For each setting, each scenario was simulated 100 times, and incidence across the time period was smoothed with a moving average. All results are reported as the median with minimum–maximum range or as a 95% uncertainty interval (UI) of values between the 2.5 and 97.5 percentiles. A population size of 100,000 was simulated to ensure constant transmission, and outputs were converted to the estimated 2018 population size. For settings with fewer than 100 reported cases, we assumed the median effect size from the simulated settings to estimate impact. Additional scenarios were simulated to account for uncertainty in parameter estimates for pre-existing adherence and assumptions of the impact of low CYP2D6 metabolism.

## Results

### Malaria epidemiology in Brazil

Due to strong case management practices, Brazil has seen a significant decrease in malaria over the last 15 years, with its lowest level reported in 2016, at 151,000 cases [4]. The number of cases has since increased, reaching 237,000 reported cases in 2018, of which more than 90% were due to *P. vivax*. Including mixed infections, 217,000 *P. vivax* cases were reported, along with 21,000 cases of *P. falciparum* (Table 1). Approximately 60% of reported cases were in males, and approximately 64% in adults aged 16 years or older. Pregnant women represented 1.5% of all malaria cases.

The age and sex distributions of cases across different transmission settings showed high levels of heterogeneity (Fig 1A). In Itaituba municipality (an archetype of occupational exposure) in Pará state, working-aged males were the main driver of cases, with an incidence of 23 cases per 1,000. In São Gabriel da Cachoeira (an archetype of peri-domestic transmission), cases were observed across all ages and equally across sex, with a higher proportion in young children. A very high incidence of 267 cases per 1,000 was observed in this setting. In peri-urban Manaus, where transmission is significantly higher than in the municipality as a whole (114 cases versus 6 cases per 1,000), a combination of occupational exposure and exposure around the household likely drove transmission dynamics.

**Table 1. Summary of 2018 malaria case reports from the national surveillance system (SIVEP-Malaria).**

| Malaria cases | *P. vivax* | *P. falciparum* | Mixed infections* |
|---|---|---|---|
| All | 216,407 (91.1%) | 20,317 (8.5%) | 912 (0.4%) |
| Males | 131,215 (91.2%) | 12,025 (8.4%) | 563 (0.4%) |
| Females | 85,192 (90.8%) | 8,292 (8.8%) | 349 (0.4%) |
| Pregnant women | 3,123 (88.3%) | 399 (11.3%) | 16 (0.5%) |
| Aged <16 years | 79,045 (92.5%) | 6,160 (7.2%) | 373 (0.4%) |
| Aged ≥16 years | 137,362 (90.3%) | 14,157 (9.3%) | 539 (0.4%) |

*Mixed infections of *P. vivax* and *P. falciparum*.

### Estimating the effective radical cure rate with chloroquine and primaquine

The current radical cure treatment regimen in Brazil for *P. vivax* cases is chloroquine 25 mg/kg total dose over 3 days and primaquine 3.5 mg/kg total dose over 7 days. The proportion of individuals eligible for primaquine depends on the underlying demographics of the population. In Fig 1B, for the baseline scenario, we estimated that among primaquine eligible cases, 66.7% fully adhered to the 7-day regimen, 8.1% were low CYP2D6 metabolisers, and primaquine had 71.3% efficacy (S1 Text). Based on these assumptions, we estimate that a median 43.7% (range 19.4%, 43.7%) of reported cases of *P. vivax* and mixed infections in 2018 were effectively treated (defined as fully adherent, with normal CYP2D6 metabolism, and with 100% clearance of parasites) under the current radical cure regimen, with some variability across different settings (Fig 1C; Table E and Fig G in S1 Text). This corresponds to a total of 92,150 effectively treated *P. vivax* cases in 2018.

### Novel chloroquine, primaquine, and tafenoquine treatment pathway

In the standard tafenoquine introduction scenario (Fig 1D), we assume the same primaquine adherence rate of 66.7% and a prevalence of low CYP2D6 metabolisers of 8.1%. Since tafenoquine is prescribed as a single dose, we assume 100% adherence. We assume that tafenoquine and primaquine have a comparable drug efficacy in the Brazilian population of 71.3%, since no significant difference was observed in the phase III clinical trial [11]. We assume no role of CYP2D6 in tafenoquine metabolism (S1 Text). Assuming the novel treatment pathway was introduced in the same epidemiological context of 2018, we estimate a median 59.8% radical cure rate (range 26.6%, 61.9%), resulting in an additional 16.1% of clinical cases being effectively treated, the equivalent of 34,950 additional cases (Table E and Fig I in S1 Text).

### Impact of tafenoquine on transmission in archetype settings

With plans for rollout in the coming years, we simulated tafenoquine introduction in Brazil in 2021 assuming similar transmission levels to those in 2018. The previously described novel treatment pathway (scenario 1 [S1]) was compared to a baseline scenario with no tafenoquine introduction and scenarios 2–6 (S2–S6) with variable age eligibility criteria, tafenoquine efficacy, and pre-existing primaquine adherence (Table 2).

Transmission intensity and the age and sex distribution of cases are major factors influencing the impact of tafenoquine. In Fig 2 and Table 3, we show the variable impact of different tafenoquine introduction scenarios (S1–S6) across the 3 archetype *P. vivax* transmission settings previously described in Fig 1.

With increasing *P. vivax* incidence, we expect a smaller effect size of tafenoquine on population-level transmission (Fig 2G). In higher transmission settings, individuals in the

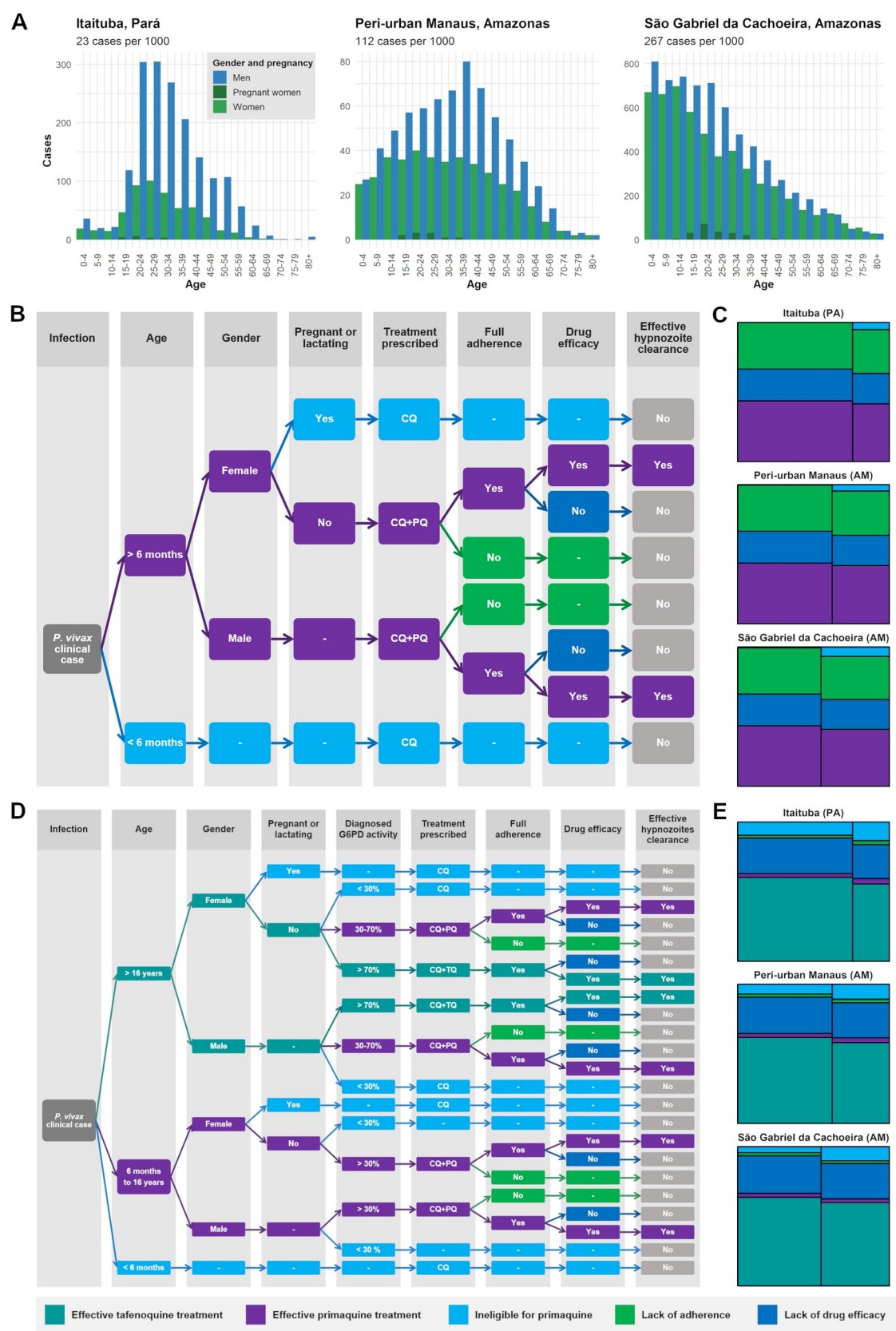

**Fig 1. Effective hypnozoite clearance with current radical cure with primaquine and with novel tafenoquine radical cure treatment pathways.** (A) The age and sex distribution of cases reported in 2018 across 3 example settings (left to right: Itaituba in Pará state [PA], peri-urban area of Manaus in Amazonas [AM], and São Gabriel da Cachoeira in Amazonas) shows the varying underlying *P. vivax* epidemiology. (B) In the treatment pathway for the current radical cure regimen with primaquine (PQ), age and sex determine eligibility for primaquine, while everyone receives chloroquine (CQ). Of those who receive PQ, some will not fully adhere to the full regimen. Of those who do fully adhere, a certain proportion will be subject to treatment failure due to low CYP2D6 metabolism and lack of drug efficacy for the given total dose. (C) We estimated that 43.1%, 42.9%, and 42.2% of cases in these 3 respective settings were effectively treated with primaquine. (D) The novel treatment pathway for radical cure with tafenoquine (TQ) introduces G6PD testing and a second age criterion for treatment eligibility. Along with infants and pregnant women, individuals with G6PD deficiency (≤30% G6PD activity) receive only CQ. Those with intermediate activity (30%–70%) and aged under 16 years are prescribed CQ and PQ. Those 16 years and older and with normal G6PD activity (≥70%) are prescribed CQ and TQ. (E) We estimated that with this novel pathway, radical cure with tafenoquine along with prescription for primaquine could achieve 59.0%, 60.5%, and 61.9% of effective radical cure rates among cases in Itaituba, Manaus, and São Gabriel da Cachoeira, respectively.

population are highly exposed to *P. vivax* parasites and develop higher levels of immunity; as a result, the majority of infections are subpatent, and only a small subset of cases will develop clinical symptoms and seek treatment (Fig J in S1 Text). For example, in Itaituba municipality in Pará state, the model predicts that 18% of infections result in clinical disease, while in São Gabriel da Cachoeira municipality in Amazonas, clinical cases represent only 8% of all infections. As a result, the impact of S1 in Itaituba is much higher than in São Gabriel da Cachoeira (31.9% versus 9.1%) since case management is able to reach a higher proportion of cases (Table 3).

The age and sex distribution of cases has a major impact on eligibility for tafenoquine and the potential impact on a population level. Introduction of a higher efficacy tafenoquine dose (S2) would improve effective radical cure rates and cause a greater reduction in transmission in highly occupational exposure settings like Itaituba, where the majority of cases are in adult males (Fig 2A and 2B). In moderate and highly peri-domestic exposure settings, where many cases are reported in those under 16 years old, the availability of a paediatric formulation of tafenoquine (S3) would be more advantageous than S2 (Fig 2C–2F). For S3 in São Gabriel da

**Table 2. Tafenoquine introduction scenarios.**

| Scenario | Primaquine pathway | | | | Tafenoquine pathway | | |
|---|---|---|---|---|---|---|---|
| | Age | G6PD activity | Adherence | Efficacy | Age | G6PD activity | Efficacy |
| Baseline | ≥6 months | No testing | 66.7% | 71.3% | | | |
| Scenario 1: Standard tafenoquine introduction | ≥6 months | 30%–70% | 66.7% | 71.3% | ≥16 years | ≥70% | 71.3% |
| Scenario 2: High-efficacy tafenoquine introduction | ≥6 months | 30%–70% | 66.7% | 71.3% | ≥16 years | ≥70% | 90.0% |
| Scenario 3: Tafenoquine introduction in children* | ≥6 months | 30%–70% | 66.7% | 71.3% | ≥2 years | ≥70% | 71.3% |
| Scenario 4: High-efficacy tafenoquine introduction in children* | ≥6 months | 30%–70% | 66.7% | 71.3% | ≥2 years | ≥70% | 90.0% |
| Scenario 5: Tafenoquine introduction with high pre-existing primaquine adherence | ≥6 months | 30%–70% | 90.0% | 71.3% | ≥16 years | ≥70% | 71.3% |
| Scenario 6: Tafenoquine introduction with low pre-existing primaquine adherence | ≥6 months | 30%–70% | 30.0% | 71.3% | ≥16 years | ≥70% | 71.3% |

Model simulations of different tafenoquine with G6PD testing strategies introduced for *P. vivax* case management in 2021 and simulated to 2030. Drug eligibility depends on the age and G6PD activity of individuals. Drug adherence and efficacy were estimated from the literature.

*It is assumed that the formulation for children is adjusted by weight to have the same efficacy as the formulation in adults.

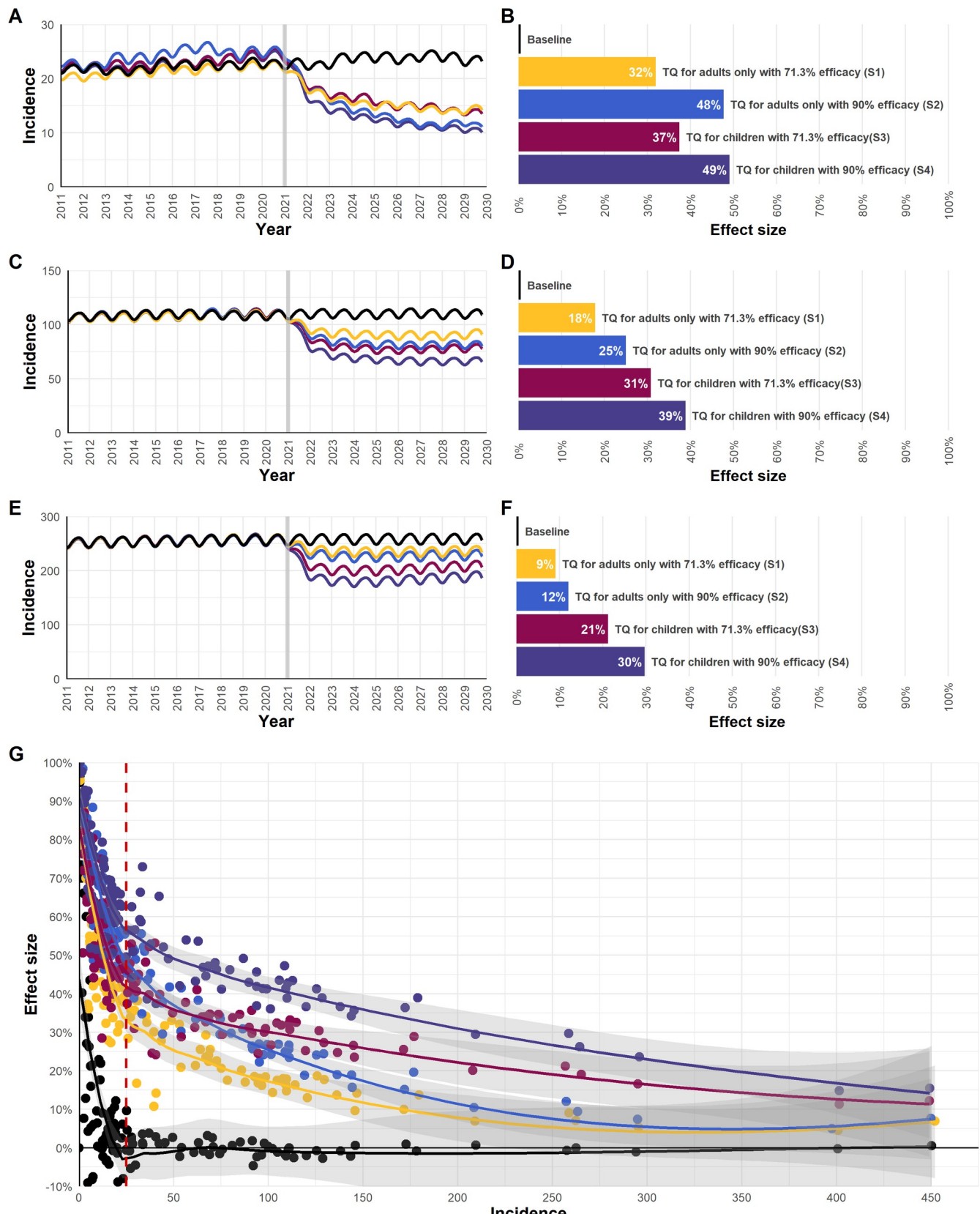

**Fig 2. Effect of introducing tafenoquine on *P. vivax* incidence in low-, moderate-, and high-transmission settings for scenarios 1–4.** (A, C, E) Model simulated *P. vivax* incidence under no intervention (baseline) and under scenarios 1–4 introducing tafenoquine with testing for G6PD deficiency in 2021 in Itaituba (A), peri-urban Manaus (C), and São Gabriel da Cachoeira (E). Incidence represents the average of 100 independent simulations per scenario, with a moving-average smoothing of the data. Incidence was calibrated to clinical cases reported in 2018 per 1,000 population. (B, D, F) The effect size (in percent) is reported as the percentage reduction in incidence after 5 years (incidence in 2025 compared to 2020) for scenarios 1–4 in Itaituba (B), peri-urban Manaus (D), and São Gabriel da Cachoeira (F). (G) The 5-year post-intervention effect size in all simulated municipalities. A Loess fitted line and 95% confidence interval bands (shading) are shown. For municipalities with an incidence below 25 cases per 1,000 population, stochastic fade out in the absence of importation prevents accurate assessment of impact.

Cachoeira, a 21.2% reduction in transmission could result in 11,600 cases averted over 5 years if children are also treated, as compared to 4,400 cases averted when treating only adults (Table 3; Table F in S1 Text).

The level of impact of tafenoquine on transmission depends greatly on the assumption of the pre-existing primaquine adherence rate (Fig K in S1 Text). In a low-transmission setting such as Itaituba, if the pre-existing adherence rate is as low as 30% (S6), tafenoquine introduction will have a significant impact on transmission, reducing incidence from 23 cases to 9.1 per 1,000, with a 5-year effect size of 60.8%. In contrast, if the pre-existing primaquine adherence is 90%, tafenoquine only has a marginal impact, reducing transmission from 23 cases to 21.7 cases per 1,000, with a 4.4% effect size. In a high-transmission setting like São Gabriel da Cachoeira, where low adherence could be an issue due to a significant indigenous population, tafenoquine could reduce transmission by a predicted 16.4%. Even such a small reduction in transmission would result in an important reduction of clinical cases, averting potentially an additional 10,100 cases over a 5-year period (Table F in S1 Text).

**Table 3. Effect of introducing tafenoquine on *P. vivax* incidence in low-, moderate-, and high-transmission settings for scenarios 1–6.**

| Modelled scenario | Municipality, state (exposure setting; cases per 1,000 population) | | | | | |
|---|---|---|---|---|---|---|
| | Itaituba, Pará (occupational; 23) | | Manaus, Amazonas (moderate occupational in peri-urban area; 114) | | São Gabriel da Cachoeira, Amazonas (peri-domestic; 267) | |
| | Incidence (95% UI) | Effect size (95% UI) | Incidence (95% UI) | Effect size (95% UI) | Incidence (95% UI) | Effect size (95% UI) |
| Scenario 1: Standard tafenoquine intervention | 14.5 (12.1 to 17.9) | 31.9% (19.8% to 44.3%) | 89.5 (86.6 to 95.0) | 17.9% (13.2% to 21.7%) | 235.9 (231.2 to 240.1) | 9.1% (6.8% to 10.4%) |
| Scenario 2: High-efficacy tafenoquine intervention | 14.0 (10.1 to 15.3) | 47.7% (34.6% to 55.1%) | 82.4 (78.4 to 84.6) | 25.0% (22.5% to 28.6%) | 226.0 (223.4 to 230.9) | 12.0% (10.4% to 13.9%) |
| Scenario 3: Tafenoquine intervention in children | 15.3 (11.6 to 17.7) | 37.4% (27.1% to 47.1%) | 76.6 (73.4 to 81.5) | 30.7% (25.2% to 34.1%) | 200.8 (197.6 to 206.5) | 21.2% (19.6% to 23.2%) |
| Scenario 4: High-efficacy tafenoquine intervention in children | 11.9 (8.6 to 14.9) | 49.1% (37.7% to 62.0%) | 67.2 (63.6 to 70.6) | 38.8% (36.7% to 42.1%) | 180.9 (178.0 to 183.4) | 29.7% (28.8% to 31.1%) |
| Scenario 5: Tafenoquine intervention with high pre-existing primaquine adherence | 21.7 (16.7 to 24.4) | 4.4% (−2.7% to 24.4%) | 104.2 (98.4 to 109.1) | 7.0% (2.2% to 12.0%) | 246.2 (242.2 to 249.3) | 5.0% (3.1% to 6.7%) |
| Scenario 6: Tafenoquine intervention with low pre-existing primaquine adherence | 9.1 (6.6 to 11.9) | 60.8% (48.0% to 68.7%) | 69.7 (65.7 to 75.1) | 35.6% (29.3% to 39.6%) | 206.9 (203.8 to 210.4) | 16.4% (15.4% to 19.2%) |

The incidence per 1,000 population and 5-year post-introduction effect size are reported for the 3 archetype settings: Itaituba, peri-urban Manaus, and São Gabriel da Cachoeira. The incidence is reported as the median of 100 independent simulations with smoothing and 95% uncertainty intervals (UIs). The effect size (in percent) is reported as the median percentage reduction in incidence 5-years after tafenoquine introduction in 2021 (incidence in 2025 compared to 2020) with 95% UI.

## National-level impact of tafenoquine

In the standard tafenoquine introduction scenario (S1), tafenoquine is expected to reduce *P. vivax* transmission regardless of the main exposure routes or transmission intensity (Fig 3) by increasing the mean effective hypnozoite clearance rate across simulated years 2021 to 2030 from a baseline of 42.3% (95% UI 40.5%–44.2%) to 62.1% (95% UI 54.2%–67.7%). By the end of the first year, we estimate a median effect size of 5.1% in transmission (Fig L and Table I in S1 Text). We expect an additional 19.9% increased impact in the second year, followed by an incremental increase until transmission stabilises over time, reaching a new equilibrium (Fig 3E). The effect size and cumulative number of averted cases will vary by setting and transmission intensity. Five years post-introduction, we estimate a median 37.5% reduction in transmission (Fig 3D). Ten low-transmission municipalities (incidence < 5 cases per 1,000) are predicted to reach over 80% reduction in transmission within 5 years post-introduction (Fig 3C). However, among moderate- and high-transmission settings (incidence > 25 cases per 1,000), we predict a median 18.2% effect size (Table J in S1 Text).

While the majority of settings will not reach elimination with S1, 4 municipalities and peri-urban Manaus would observe over 8,000 averted cases over this period. An estimated 177,000 new *P. vivax* cases would be averted during the first 5 years of tafenoquine rollout among simulated settings (Fig 3F; Table H in S1 Text). Among settings reporting fewer than 100 cases in 2018 (i.e., settings that were not simulated), a 37.5% median effect size over 5 years could result in an additional 36,800 averted cases assuming no change in annual reported cases (Table H in S1 Text). Overall, over 214,000 *P. vivax* cases would be averted across all endemic settings in S1 (Table H in S1 Text).

We assessed the impact of alternative strategies, including a paediatric formulation of tafenoquine and a safely administered high-efficacy dose across all settings (S2–S4). Improving tafenoquine efficacy to 90% in adults (S2) would have a greater overall impact on transmission than introducing a paediatric formulation of the standard efficacy dose in children ages 2 to <16 (S3) (Fig 3D). Compared to S1, S2 would improve impact by an additional 14.0% within 5 years, as compared to 7.5% for S3, since larger majorities of settings have cases in adult males due to occupational exposure (Table 4). On the other hand, S3 would result in more cases averted due to a higher burden among children in high-transmission settings being able to benefit from radical cure with tafenoquine. Having a safe high-efficacy dose for all ages (S4) would have the most benefit, reducing transmission by an additional 22.1% compared to S1 (median effect size of 59.6%) and averting twice as many cases (Table 4). Such a scenario would help reach an 80% reduction in *P. vivax* transmission in 18 simulated municipalities (mainly those with low transmission) and reach elimination in 4 municipalities. The vast majority of municipalities across S1–S4 would not reach elimination with improved case management alone.

Adherence to primaquine can be highly variable across different populations worldwide; therefore, we assessed the impact of different levels of pre-existing primaquine adherence on the effect of introducing tafenoquine. The 37.5% median impact of S1 relied heavily on the assumption of a 66.7% pre-existing primaquine adherence rate in the Brazilian population. If the pre-existing primaquine adherence rate is higher, at 90% (S5), tafenoquine only contributes a 17.7% reduction over 5 years. However, if the pre-existing adherence rate is as low as 30% (S6), tafenoquine introduction has a substantial impact on transmission, with a median effect size of 66.8% (Fig 3D; Table 4).

The total number of treatment courses of chloroquine, primaquine, and tafenoquine and the number of G6PD tests administered over a 5-year post-intervention period were estimated from model outputs (Table 4). As transmission decreases overtime, fewer chloroquine doses

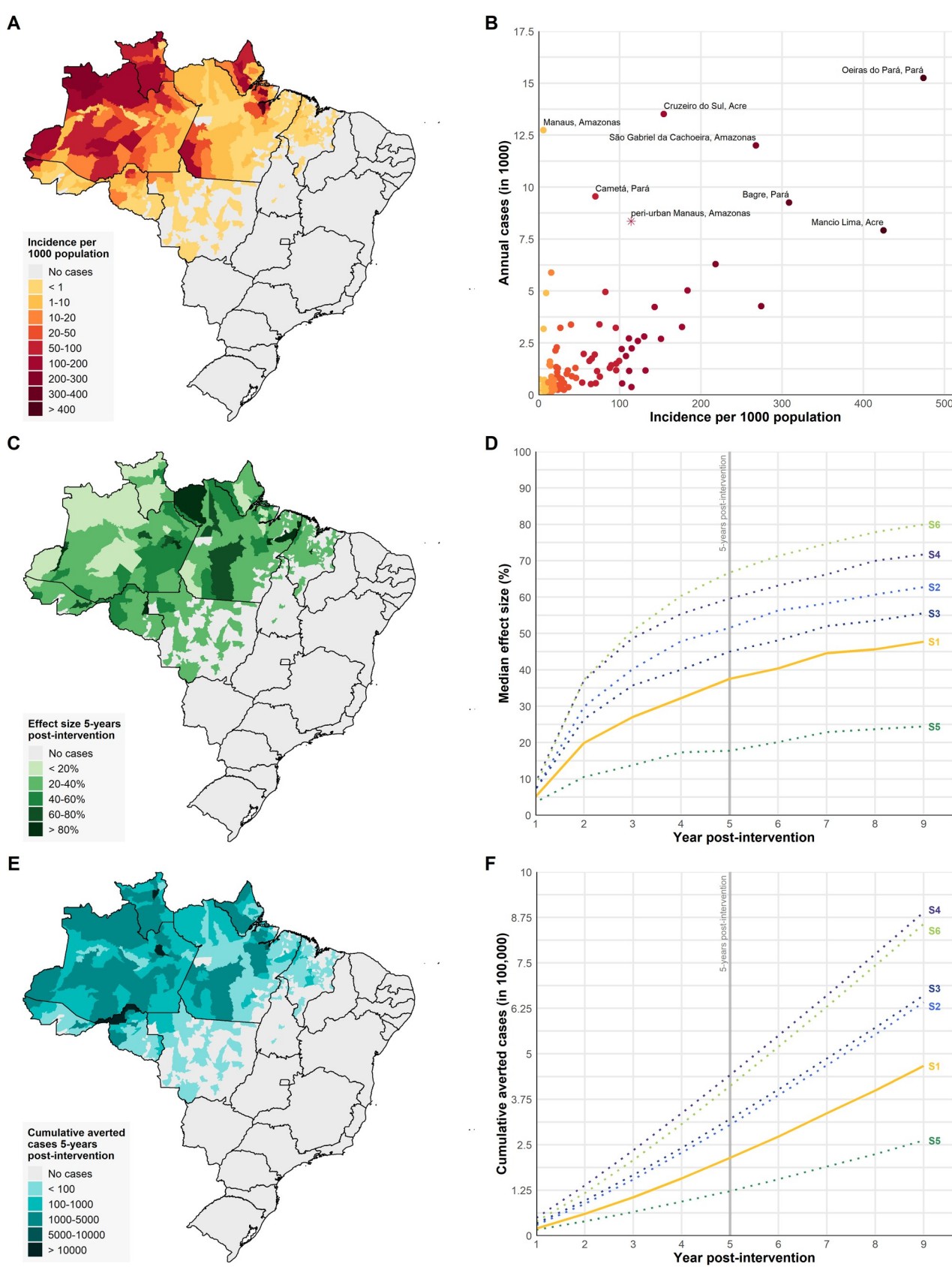

**Fig 3. Effect of introducing tafenoquine across the *P. vivax* endemic region of Brazil.** (A) *P. vivax* incidence per 1,000 population across all municipalities (*n* = 424) reporting at least 1 case in 2018. (B) Comparison of incidence and total cases reported in 2018, with selected municipalities labelled (peri-urban Manaus setting shown as star). (C) The effect size (percent change in incidence) 5 years after introduction of the standard tafenoquine introduction scenario (scenario 1). (D) The median effect size across all municipalities over time for scenarios 1–6 (S1–S6). S1 is shown as a solid line and S2–S6 are shown as dotted lines. (E) The total cumulative number of averted cases reached 5 years post-introduction for S1 as compared to the baseline scenario for all 424 municipalities. (F) The total cumulative number of averted cases over time for all municipalities for S1–S6. Maps generated with shapefiles from the R package malariaAtlas.

would need to be prescribed overall. At least 846,000 G6PD tests will need to be delivered over this period in S1. Even after the introduction of tafenoquine, more primaquine courses will be prescribed than tafenoquine courses due to the demography of the case population.

## Other contributing factors

Another alternative strategy to improving radical cure among those ineligible for tafenoquine (children or those with intermediate G6PD activity) is switching from the low (3.5 mg/kg total dose) to high (5 mg/kg total dose) primaquine regimen. Switching to higher efficacy tafenoquine and primaquine (S7 in Table J in S1 Text) as compared to only high-efficacy tafenoquine (S2) would improve the 5-year median effect size by 4.0% (from 51.5% to 55.5%), especially in settings with more cases in children who would benefit from improved primaquine efficacy (Table J and Fig M in S1 Text).

Cytochrome P450 2D6 (CYP2D6) is an important enzyme in the activation of drugs in the liver and in particular of 8-aminoquinoline [17]. If low CYP2D6 metabolism also compromises tafenoquine efficacy (S8 in Table J in S1 Text), we predict a reduced impact (from 37.5% to 28.6%) (Table J and Fig N in S1 Text). A lower prevalence of low CYP2D6 metabolism of 4% (S9 in Table J in S1 Text), compared to 8.1%, would not have a significant impact. However, a higher prevalence of 20% (S10 in Table J in S1 Text) would compromise 8-aminoquinoline efficacy more significantly.

**Table 4. Effect of introducing tafenoquine across the *P. vivax* endemic region of Brazil.**

| Modelled scenario | Effect size (95% UI) | Cases averted | Total courses | | | G6PD tests | Percent effective radical cure (95% UI) |
|---|---|---|---|---|---|---|---|
| | | | CQ | PQ | TQ | | |
| Baseline | | | 1,058,200 | 1,022,000 | 0 | 0 | 42.3% (40.5% to 44.2%) |
| Scenario 1: Standard tafenoquine intervention | 37.5% (7% to 99.4%) | 177,600 | 881,400 | 411,100 | 341,100 | 846,900 | 62.1% (54.2% to 67.7%) |
| Scenario 2: High-efficacy tafenoquine intervention | 51.5% (9.6% to 98.3%) | 254,100 | 804,900 | 395,000 | 288,900 | 772,400 | 72.7% (58.7% to 81.3%) |
| Scenario 3: Tafenoquine intervention in children | 45.0% (19.2% to 89.8%) | 276,700 | 782,400 | 95,200 | 565,500 | 749,300 | 68.6% (67.3% to 70.1%) |
| Scenario 4: High-efficacy tafenoquine intervention in children | 59.6% (26.6% to 99.7%) | 383,600 | 675,500 | 87,500 | 476,500 | 645,100 | 84.9% (82.9% to 86.3%) |
| Scenario 5: Tafenoquine intervention with high pre-existing primaquine adherence | 17.7% (−0.7% to 98.1%) | 105,100 | 953,900 | 423,600 | 387,200 | 915,600 | 68.3% (65.5% to 71.6%) |
| Scenario 6: Tafenoquine intervention with low pre-existing primaquine adherence | 66.8% (15.4% to 97.1%) | 345,900 | 713,200 | 371,100 | 243,800 | 685,600 | 49.8% (34.5% to 60.1%) |

The median effect sizes 5 years after tafenoquine introduction in 2021 for scenarios 1–6 are reported across 126 simulated municipalities (Manaus includes only peri-urban Manaus). The cumulative averted cases are compared to the baseline scenario. The total numbers of chloroquine (CQ), primaquine (PQ), and tafenoquine (TQ) courses and G6PD tests are reported as a cumulative sum from the period 2021 to 2025. The mean effective radical cure rate is reported as the proportion of primaquine and tafenoquine doses that completely clear hypnozoites compared to the total doses prescribed during the 2021 to 2030 period. Issues of adherence and drug efficacy are accounted for in the model. Cases and dose values are rounded to the nearest 1,000.

## Discussion

Introducing a novel radical cure regimen to effectively clear *P. vivax* hypnozoites with chloroquine and a single dose of tafenoquine with G6PD testing is predicted to have a significant impact on transmission over time. Currently in Brazil, radical cure with primaquine and chloroquine without G6PD testing is estimated to provide effective radical cure to less than half of symptomatic cases. The major advantage of introducing tafenoquine treatment is full compliance with the single-dose treatment compared to a longer 7-day primaquine regimen, resulting in an increase of the overall rate of effective radical cure. In addition, tafenoquine provides an extended prophylactic period protecting individuals from new infection. Furthermore, preventing future relapses will prevent onward transmission to mosquitoes; thus, introducing tafenoquine will lead to reduction in *P. vivax* transmission on the population level. Notably, these benefits will even extend to individuals who do not receive tafenoquine. By accounting for these non-linear dynamics of malaria transmission, mathematical models estimate the full public health impact of improved case management.

Our model predictions show that increasing the effective radical cure rate from 42% to 62% could achieve a median 38% reduction in transmission within 5 years across the *P. vivax* endemic region of Brazil. Over this period, such a reduction in transmission is predicted to avert over 214,000 cumulative cases compared to the baseline scenario. The potential size of the impact of tafenoquine rollout on transmission will vary greatly depending upon several epidemiological characteristics and the rollout strategies put in place. Notably we predict that *P. vivax* elimination will not be achieved in the majority of transmission settings by improving case management among clinical cases alone.

Our analysis provides support for continuing the development of paediatric formulations and higher dose formulations of tafenoquine. A paediatric formulation that reduces the minimum age for tafenoquine treatment from 16 years to 2 years is estimated to cause larger reductions in transmission, with the median effect size increasing from 38% to 45%. A higher dose formulation that could be safely administered to provide higher efficacy would have even greater impact, with an effect size on transmission of 51%. Combining safe paediatric and high-dose formulations would further increase the effect size to 60%. This large potential public health impact indicates that further trials on tafenoquine formulations should be prioritised, in both Brazil and the rest of the *P. vivax* endemic world.

The age distribution of cases and their eligibility for tafenoquine will have a substantial impact on tafenoquine rollout. In Brazil, children under 16 years account for 34% of *P. vivax* cases, and varying levels of peri-domestic and occupational exposure create heterogeneous age-sex structure of cases. In populations where a high proportion of cases are in adults ($\geq$16 years of age), tafenoquine is likely to be more effective.

The impact of introducing tafenoquine into case management practices of *P. vivax* will vary based on the transmission intensity. Notably, the effect size is largest in low-transmission settings (incidence $<$ 25 per 1,000), with simulations achieving elimination in some instances. However, due to the low prevalence of *P. vivax*, there is greater uncertainty in these estimates due to the stochastic nature of the model, and caution should be taken when interpreting these results. These settings may also be subject to sustained transmission due to importations, which were not accounted for in the model. One limitation of modelling these scenarios at the municipality level is that underlying heterogeneities in transmission may be masked. Several municipalities cover a large spatial area that can encompass sparse hotspots that may vary greatly in terms of transmission between urban, rural, and worksite contexts [30–32]. For instance, in the municipality of Manaus, the overall municipal-level incidence per 1,000 is

estimated at 6 cases, while in the peri-urban area, it is estimated at 114 cases. Therefore, care should be taken when extrapolating these results to intra-municipal communities.

In populations with high levels of transmission, we expect that a large number of asymptomatic infections will contribute to onwards transmission. Asymptomatic infections prevent treatment-seeking behaviour of those who may potentially benefit from tafenoquine. At higher transmission intensities, the effect size of tafenoquine introduction is estimated to be significantly less; however, even a small relative effect size may result in large numbers of averted cases over time. High-transmission pockets will require additional intervention strategies to tafenoquine to reach pre-elimination status. Vector control, reactive case management, and identification of asymptomatic individuals with serological tools should be considered alongside improved case management.

Primaquine coverage and adherence are major determinants of the size of the effect that tafenoquine will have on transmission. Brazil, unlike other *P. vivax* endemic regions, provides high coverage of primaquine to clinical cases, and the handful of studies providing estimates of adherence to the 7-day regimen in South America show generally high levels of adherence. This may not be the case in other endemic regions where national guidelines require G6PD testing before providing primaquine or recommend a 14-day regimen, which can result in very little coverage and potentially very low adherence.

Even in Brazil, where primaquine coverage is high, estimating primaquine adherence is challenging, and adherence can also have a high degree of variability across different communities [24,25,33]. The model assumption of the pre-existing primaquine adherence rate significantly affected the estimated impact of tafenoquine rollout. In populations with low pre-existing levels of primaquine adherence, the incremental benefit of introducing tafenoquine is likely to be greater. Groups that have been anecdotally reported to have low primaquine adherence include indigenous populations and males working in forestry and mining. Conversely, in populations with high primaquine adherence (for example, if specific activities are implemented to enhance primaquine adherence), the incremental benefit of introducing tafenoquine is likely to be less. While in Brazil high primaquine coverage with varying adherence could have large impacts on tafenoquine rollout, we also expect such high levels of variability in settings with very low primaquine coverage. Countries considering the potential benefit of adopting tafenoquine are recommended to assess primaquine coverage and adherence across different communities, as few data are available [10].

The availability of G6PD activity testing to protect against haemolytic events in G6PDd individuals will serve as the first major gateway to introduction of tafenoquine. G6PDd prevalence and the performance of the G6PD diagnostic tests will be critical to determining eligibility for 8-aminoquinolines. With G6PD testing, a higher proportion of cases will become ineligible for radical cure for 2 reasons. First, G6PD testing will protect G6PDd individuals from 8-aminoquinoline-induced haemolysis with perfect sensitivity using the SD Biosensor G6PD test [34]. Second, the test may misclassify individuals with intermediate G6PD activity to deficient status, thus reducing the number of individuals prescribed primaquine and compromising impact on transmission. More importantly, we predict that expanding access to tafenoquine treatment with G6PD testing will lead to an increase in the number of G6PD tests administered and courses of drugs prescribed in the short term. As effective implementation will reduce population-level *P. vivax* transmission in the long term, it will also lead to an overall reduction in the number of individuals requiring *P. vivax* treatment and G6PD tests. One safety concern that may arise is that 20% of these individuals with intermediate G6PD activity may also be misclassified into the normal activity group, resulting in severe haemolytic events due to tafenoquine [35]. Further studies on the

potential effect of G6PD diagnostics and 8-aminoquinoline on patient safety and associated health costs should be pursued.

Estimating effective radical cure relies on assumptions of 8-aminoquinoline efficacy. Our baseline tafenoquine rollout scenario assumed equal efficacy between primaquine and tafenoquine with chloroquine based on evidence demonstrated by the phase III clinical trial [11,20]. This may vary for other contexts due to the dosing schedule and duration of prophylaxis of the blood-stage drug prescribed, intrinsic characteristics of local parasite species, and the prevalence of low CYP2D6 metabolism in the population. Along with expanding tafenoquine administration to all age groups, improving 8-aminoquinoline efficacy would accelerate the path to lower *P. vivax* transmission levels. The main concern of identifying a higher efficacy tafenoquine dose is patient safety. Since G6PD testing will become standard, stricter cutoffs may be considered for those receiving such a dose. In addition, switching from a lower primaquine dosing regimen among those with intermediate G6PD activity to a higher regimen may also improve 8-aminoquinoline efficacy while ensuring patient safety.

Our study had several limitations. First, the model did not account for importations, and municipality-level incidence was calibrated to reported clinical cases, which may not accurately reflect community-level transmission and the contribution of asymptomatic cases. Second, while treatment efficacy behaves on a continuous scale, we treated drug efficacy and low CYP2D6 metabolism as binary conditions and did not account for inter-individual heterogeneity [36]. Since the settings we modelled were calibrated to 2018 *P. vivax* incidence, we assumed constant transmission up to 2021, when tafenoquine is introduced. Therefore, we did not account for dynamic changes in incidence informed by historical trends. We also did not account for the possibility of potential health system disruptions due to COVID-19. Drawing from recently published work that has modelled the impact of COVID-19 on malaria transmission in endemic settings in Africa [37], we would expect some level of disruption to malaria case management and thus an increase in incidence. A thorough assessment of the impact of COVID-19 on malaria transmission, accounting for variability in seasonal trends of *P. vivax* as well as delayed introduction of the new treatment scheme, is required.

Despite the clear benefits of improving radical cure treatment and access, one should not expect tafenoquine alone to lead to *P. vivax* elimination. Even in Brazil, where case management practices with primaquine have had significant impact on prevalence and tafenoquine will be introduced within the next few years, elimination objectives for 2030 will not be met with improved case management alone. Variability across transmission settings requires tailored intervention strategies adapted to the local context to achieve maximum impact. In order for *P. vivax* endemic countries to reach 2030 elimination goals, additional commitments to scale up treatment coverage, vector control, and diagnostics in parallel with improved radical cure will help localities reach pre-elimination and elimination status.

## Supporting information

**S1 Text. Supplementary information for "Estimated impact of tafenoquine for *Plasmodium vivax* control and elimination in Brazil: A modelling study" by Nekkab et al.**
(PDF)

## Acknowledgments

The authors would like thank Penny Grewal Daumerie, Elodie Jambert, Melanie Larson, Gonzalo Domingo, and Alexandre Menezes for their valued contributions, support, and fruitful discussions.

## Author Contributions

**Conceptualization:** Daniel Villela, Ivo Mueller, Michael White.

**Data curation:** Narimane Nekkab, Raquel Lana, Wuelton Monteiro.

**Formal analysis:** Narimane Nekkab.

**Funding acquisition:** Daniel Villela, Michael White.

**Investigation:** Narimane Nekkab.

**Methodology:** Narimane Nekkab, Michael White.

**Project administration:** Ivo Mueller, Michael White.

**Resources:** Thomas Obadia, Michael White.

**Software:** Thomas Obadia, Michael White.

**Supervision:** Ivo Mueller, Michael White.

**Validation:** Raquel Lana, Marcus Lacerda, Thomas Obadia, André Siqueira, Wuelton Monteiro, Daniel Villela, Ivo Mueller, Michael White.

**Visualization:** Narimane Nekkab.

**Writing – original draft:** Narimane Nekkab, Ivo Mueller, Michael White.

**Writing – review & editing:** Narimane Nekkab, Raquel Lana, Marcus Lacerda, Thomas Obadia, André Siqueira, Wuelton Monteiro, Daniel Villela, Ivo Mueller, Michael White.

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
