## [Editor Report · Decision Letter 0]

5 Aug 2020

Dear Dr Nekkab, 

Thank you for submitting your manuscript entitled "Tafenoquine for Plasmodium vivax control and elimination: a modelling case study of Brazil" for consideration by PLOS Medicine.

Your manuscript has now been evaluated by the PLOS Medicine editorial staff as well as by an academic editor with relevant expertise and I am writing to let you know that we would like to send your submission out for external peer review.

Kind regards,

Artur Arikainen,

Associate Editor

PLOS Medicine

---

## [Decision Letter · Decision Letter 1]

14 Oct 2020

Dear Dr. Nekkab,

Thank you very much for submitting your manuscript "Tafenoquine for Plasmodium vivax control and elimination: a modelling case study of Brazil" (PMEDICINE-D-20-03573R1) for consideration at PLOS Medicine. 

Your paper was evaluated by a senior editor and discussed among all the editors here. It was also discussed with an academic editor with relevant expertise, and sent to independent reviewers, including a statistical reviewer (#3). The reviews are appended at the bottom of this email and any accompanying reviewer attachments can be seen via the link below:

[LINK]

In light of these reviews, I am afraid that we will not be able to accept the manuscript for publication in the journal in its current form, but we would like to consider a revised version that addresses the reviewers' and editors' comments. Obviously we cannot make any decision about publication until we have seen the revised manuscript and your response, and we plan to seek re-review by one or more of the reviewers. 

We expect to receive your revised manuscript by Nov 04 2020 11:59PM. Please email us (plosmedicine@plos.org) if you have any questions or concerns.

We look forward to receiving your revised manuscript. 

Sincerely,

Emma Veitch, PhD

PLOS Medicine

On behalf of Artur Arikainen, PhD, Associate Editor, 

PLOS Medicine

plosmedicine.org

*In the last sentence of the Abstract Methods and Findings section, please add a sentence/note about any key limitation(s) of the study's methodology.

*Please note the author guidelines for the "Author summary" structure - https://journals.plos.org/plosmedicine/s/revising-your-manuscript#loc-author-summary - each subsection should be formatted as 2-3 single sentence bullet points for each of the sections (ie not paragraph text). See here for an example - https://journals.plos.org/plosmedicine/article?id=10.1371/journal.pmed.1003318

*On page 7, the methods note the SIVEP database for certain parameter estimates (and this is also noted in the supplementary material pdf) - can a citation, web link or reference be added for the database so that readers can ascertain where the underlying data can be found? The same is true for other databases noted as the source for parameters fed into the model.

*On page 8, the paper notes the data source for a point about intra-municipality heterogeneity in transmission with reference to "Monteiro et al., manuscript in preparation". If that paper (the Monteiro one) isn't published by the time that the current PLOS Medicine submission might be ready for acceptance, assuming it fully passes editorial and peer review, then the unpublished paper can't really be cited as the support for a particular claim or assumption - because readers will not be able to go and check it. We realise this is challenging, but the authors will need to cite some published or accepted article or database as the source for the claim. (see below). If this isn't possible then it is possible to convert the intext note to a "personal communication" instead (and as stated in our author guidelines the authors need to be prepared to support, if requested, the pers.comm. with a letter from the relevant individual). 

******

https://journals.plos.org/plosmedicine/s/submission-guidelines#loc-reference-style - 

Do not cite the following sources in the reference list:

>Unavailable and unpublished work, including manuscripts that have been submitted but not yet accepted (e.g., “unpublished work,” “data not shown”). Instead, include those data as supplementary material or deposit the data in a publicly available database.

>Personal communications (these should be supported by a letter from the relevant authors but not included in the reference list)

******

*If possible please reformat the citation style into PLOS Medicine's format (should be straight forward if using referencing software) - this should use callouts formatted as sequential numerals in square brackets (not superscript). Many thanks

*Please clarify in the methods section whether the analytical approach reported in the paper corresponds to one laid out in a prospective protocol or analysis plan. Please state this (either way) early in the Methods section.

*Please include in the Discussion section a more explicit (ie signposted) description of key limitations (and strengths) of the model, specifically noting any which might preclude drawing firm conclusions or make the key findings less secure. At the moment a number of issues of interpretation are discussed but none are explicitly described as limitations of the model per se. 

Comments from the reviewers:

Reviewer #1: This is an important contribution on the long journey to malaria elimination in Latin America which will require tafenoquine. Most modelling papers are criticized for being overly optimistic which is not a feature of this manuscript; it clearly claims 'no silver bullet' against malaria. The importance of getting a pediatric formulation and dealing more with adult males (likely miners in the Brazilian context) is certainly made in the discussion. Although G6PD deficiency is clearly discussed, it is not belaboured which is good. One additional point could be considered is that all malaria programs require a working health system. Brazil's health system has been massively disrupted by COVID-19 and although better than Venezuela, that is not saying much in the American context. Models are based on what has happened up to a point, some mention that things can also go backwards when health systems are disrupted may need to be inserted.

I am neither a modeler nor a statistician so although I see no errors of fact or interpretation, would hope that the editor has brought in other reviewers with such skills to examine the numbers.

Reviewer #2: I enjoyed reading this carefully written paper. The authors have modeled the impact of tafenoquine deployment on P. vivax malaria in Brazil. The underlying model is well thought through and the assumptions have been explained more than adequately. The main result - TQ will help reduce transmission but is not a silver bullet - is very important.

In the interests of reproducibility, could the authors make their code available?

My only other comments concern the epistemic uncertainty around some of the modelling assumptions, and these are mainly cosmetic and should not impact the main results of the paper. All the comments pertain to the supplementary text, which is well worth reading!

G6PD deficiency: this is a nice analysis. It might be worth pointing out that the enzyme activity data (Fig S4) are all from countries with G6PD mutations that are thought to be more severe (i.e. lower activity in deficient red cells) than the A-202 mutation that is the major mutation in South America: Bangladesh has Orissa, Kalyan-Kerala and Mahidol; Cambodia has Viangchan; Indonesia has mostly Vanua Lava; Israel has Mediterranean. In particular Vanua Lava and Med are probably the most severe variants found at >1% allele frequency. Therefore your estimate of the number of heterozygous females below the 70% threshold is probably conservative (overestimate). It's also possibly worth mentioning that the data from Pal et al (African Americans) are directly applicable to Brazil as they will be most certainly A-202.

CYP2D6: I think that this section has the highest epistemic uncertainty and it would be useful to mention this. You rightly note "The association between CYP2D6 activity and the efficacy of 8-aminoquinolines against P. vivax hypnozoites is undoubtedly a complex one, with multiple sources of evidence combining to suggest a non-linear dose-response relationship between activity and efficacy."

I want to add a few comments on this complex picture.

1. Tafenoquine is not affected by CYP2D6: the oft quoted results from St Jean et al are unfortunately confounded. Their Fig 1 shows in the primaquine group a higher relapse frequency in intermediate metabolisers as compared to extensive metabolisers, but the opposite pattern in the tafenoquine group. But this result is meaningless because both relapse patterns and distribution of CYP2D6 is highly dependent on study country (DAG: relapse <- country -> CYP2D6)! I can't work out what the adjusted result would be without the raw data (only marginal distributions are given in the paper), but the effect could be the opposite (Simpson's paradox). Lacerda et al (NEJM) does not appear to adjust for country when analyzing the effect of CYP2D6 on primaquine recurrence.

2. Primaquine is affected by CYP2D6: the results from Kevin Baird and colleagues are very compelling. However, in our analysis of primaquine efficacy in vivax patients along the Thai-Myanmar border (Taylor et al, Nature Communications), we estimated a very low primaquine failure ratio (~3%). There are lots of intermediate metabolisers (AS=0.5 or 1) in this population. Unfortunately, we didn't put this in the paper, so you can't cite it, but for the record, out of 349 patients who were genotyped, 5 had an AS score = 0, 40 had AS=0.5; 90 had AS=1. There was no association between the AS score and estimated probability of primaquine failure. This may be due to a dose effect as these patients were given high dose primaquine. it may be due to the epidemiology or other factors we don't know. It also may be due to the scoring system - see next point.

3. AS scores: this is a scoring system used by pretty much everyone in order to map genotype to phenotype. It's an incredibly useful because the phenotype test is very labor intensive. However, it has many issues. First there is large inter-individual variability for the same genotype (Chiba et al, Inter-individual variability of in vivo CYP2D6 activity in different genotypes 2012). Second, the mapping might not be very accurate for some of the genotypes. From the inventors of the system themselves (Andrea Gaedigk et al. Ten Years' Experience with the CYP2D6 Activity Score: A Perspective on Future Investigations to Improve Clinical Predictions for Precision Therapeutics. 2018):

"[..] the classification of alleles as increased, normal, decreased or no function is crude, and does not take into account substrate-dependent effects of an allelic variant toward various CYP2D6 substrates [7]. This challenge is perhaps best exemplified by CYP2D6*10, a decreased function allele, which has been associated with substantially reduced levels of activity. [..] the current assignment of a value of 0.5 to this allele classifies homozygous CYP2D6*10/*10 patients as NM in CPIC guidelines, which may not be appropriate for all drugs. A literature review and assessment performed in 2013 [8] did not produce sufficient evidence to recommend a reclassification of the CYP2D6*10 allele by assigning a lower value of 0.25 to better reflect the level of reduction of this allele at that time. However, a recent extensive literature review and assessment performed for the CYP2D6/tamoxifen gene/drug pair CPIC guideline produced strong evidence for a separate recommendation for CYP2D6*10 containing genotypes [9]"

If I understand correctly, *10 constitutes about half the poor/intermediate metabolisers in Brazil? It's probably fair to say that there is considerable uncertainty as to the effect of CYP2D6 on the use of both tafenoquine or primaquine.

Minor comments:

* Is one decimal point necessary in the abstract results?

* Line 40: for P. vivax

* Line 92: suggest changing this to "demonstrated non-inferiority with respect to low-dose primaquine to prevent relapse"

* Line 97: X-linked

James Watson

Reviewer #3:

It would be useful to discuss the selection of a Susceptible Infectious Susceptible model as opposed to a Susceptible Infectious Recovered one, say an extension of the McDonald or May-type, as often used in modelling Malaria

This is important since the long term dynamics of the different types of model are different

Also, given that malaria transmission is mostly local (at least within the 126 municipalities), did the authors consider a structured, say meta-population-type, model?

That would differentiate the stochastic and the deterministic models substantially, including their mean behaviour

In relation to the model type, treatment efficacy and prevalence, are often assumed to be exchangeable, as opposed to identical, across regions/municipalities. This assumption is typically adopted in order to allow for similarity and heterogeneity simultaneously. Would that be an option in this paper?

Such an approach would incorporate Manaus-type heterogeneity in a natural manner

Please add the details of the algorithm used for fitting the Bayesian mixture model. In particular, what was the approach used for dealing with label switching?

Are the mosquito population dynamics essentially absent from the model?

Is there an implicit assumption that they remain the same or stationary over time? Please discuss

The 37% at risk assumption (supplement page 4) seems important but is informed by a future manuscript. Please give in the present paper a short description of the data and methods resulting in this number

How extensive is the evidence, especially long term, on Tafenoquine? What is the best way to incorporate the associated uncertainty? Please discuss as this is a crucial component of the model and partly drives the results and conclusions. 

[LINK]

---

## [Decision Letter · Decision Letter 2]

16 Dec 2020

Dear Dr. Nekkab,

Thank you very much for re-submitting your manuscript "Tafenoquine for Plasmodium vivax control and elimination: a modelling case study of Brazil" (PMEDICINE-D-20-03573R2) for review by PLOS Medicine.

I have discussed the paper with my colleagues and the academic editor and it was also seen again by two reviewers. I am pleased to say that provided the remaining editorial and production issues are dealt with we are planning to accept the paper for publication in the journal.

[LINK]

We look forward to receiving the revised manuscript by Dec 23 2020 11:59PM.   

Sincerely,

Artur Arikainen

Associate Editor

PLOS Medicine

plosmedicine.org

Requests from Editors:

1. Title: Please update to: “Estimated impact of tafenoquine for Plasmodium vivax control and elimination in Brazil: A modelling study”

2. Please update your Competing Interests statement on the submission form to say: “The authors have declared that no competing interests exist.”

3. Data Availability Statement: You state in our first question on the submission form that restrictions apply, then in the second question that all data are in the manuscript and its files. Please confirm which situation is applicable. 

a. If the data are freely or publicly available, note this and state the location of the data: within the paper, in Supporting Information files, or in a public repository (include the DOI or accession number). 

b. If the data are owned by a third party but freely available upon request, please note this and state the owner of the data set and contact information for data requests (web or email address). Note that a study author cannot be the contact person for the data.

c. If the data are not freely available, please describe briefly the ethical, legal, or contractual restriction that prevents you from sharing it. Please also include an appropriate contact (web or email address) for inquiries (again, this cannot be a study author).

4. Abstract:

a. Please add a sentence at the start to mention, eg. the total number of infections in 2018, and the proportion of vivax infections.

b. Line 25: Please specify the dates for data collected.

c. Line 31: Replace “expected” with “estimated”.

d. Please quantify your results with 95% CIs and p values.

e. Please give some additional details, e.g., the modelled year of 2021; relevant 5 year window; observed regimen adherence and assumed adherence to tafenoquine.

f. Line 37: Please clarify that this is “case importations”?

g. Conclusions: Begin with: “In our modelling study, we predicted that…”

h. Line 42 (and later 518): Please replace “silver bullet” with a different term, to avoid terms all readers may not be familiar with.

5. Author Summary: 

a. Overall, we would recommend shortening your Author Summary by 1-2 bullet points per section. Please also adjust the tone to be more suitable to a lay reader, and avoid too much technical language, eg. “haemolytic events”, “G6PDd” (unless you define it first), “phenotypic activity”, “CYP2D6 metabolisation”, “peri-domestic”.

b. Line 51 (and throughout, eg line 100): Please remove brand names.

c. Line 69: Please say “first, to our knowledge” or similar.

d. Line 77: Please delete: “Even in a setting such as Brazil with strong pre-existing case management with primaquine, we would not expect tafenoquine to be the silver bullet that will lead to P. vivax elimination.”

6. Please move your citations to before punctuation and remove spaces within the citation, eg: “…strategy tackling P. vivax malaria [7,8].”

7. Methods: 

a. Line 161: Please give exact date ranges for the case reports used.

b. Please state that separate ethical approval was not required for your study.

8. The terms gender and sex are not interchangeable (as discussed in http://www.who.int/gender/whatisgender/en/ ); please use the appropriate term.

9. Please provide a URL or DOI for reference 23, and spell out the full author organisation name.

10. Please check that journal short names are presented consistently in your reference list, eg. should be “PLoS Med”.

Comments from Reviewers:

Reviewer #2: The authors have answered all queries satisfactorily. I congratulate them on an excellent piece of work!

Reviewer #3: The authors have revised their paper and this now represents an improved manuscript, acceptable for publication

[LINK]

---

## [Editor Report · Decision Letter 3]

6 Jan 2021

Dear Dr Nekkab, 

On behalf of my colleagues and the Academic Editor, Lorenz von Seidlein, I am pleased to inform you that we have agreed to publish your manuscript "Estimated impact of tafenoquine for Plasmodium vivax control and elimination in Brazil: A modelling study" (PMEDICINE-D-20-03573R3) in PLOS Medicine.

PRESS

Sincerely, 

Artur A. Arikainen 

Associate Editor 

PLOS Medicine